# SBSC: Step-by-Step Coding for Improving Mathematical Olympiad Performance

**Kunal Singh**[*]   **Ankan Biswas**   **Sayandeep Bhowmick**   **Pradeep Moturi**

Fractal AI Research

Mumbai

## Abstract

We propose Step-by-Step Coding (SBSC): a multi-turn math reasoning framework that enables Large Language Models (LLMs) to generate sequence of programs for solving Olympiad level math problems. After each turn/step, by leveraging the code execution outputs and programs of previous steps, the model generates the next sub-task and the corresponding program to complete it. SBSC allows more granular, flexible and precise approach to problem-solving compared to existing methods. Extensive experiments highlight the effectiveness of SBSC in tackling competition and Olympiad-level math problems. For Claude-3.5-Sonnet, we observe SBSC (greedy decoding) surpasses existing state-of-the-art (SOTA) program generation based reasoning strategies by absolute 10.7% on AMC12, 8% on AIME and 12.6% on MathOdyssey. Given SBSC is multi-turn in nature, we also benchmark SBSC's greedy decoding against self-consistency decoding results of existing SOTA math reasoning strategies and observe performance gain by absolute 6.2% on AMC, 6.7% on AIME and 7.4% on MathOdyssey. Scripts & Data is uploaded at this link for reproducibility.

## 1   Introduction

Mathematics is considered as a critical benchmark to measure the reasoning abilities of the Large Language Models (LLMs) [5, 8, 1, 23, 2, 21] due to the complex and creative nature of the subject. The current generation of advanced LLMs, GPT-4o [1], Claude-3.5-Sonnet [2], Gemini-ultra [23] have achieved high scores on elementary GSM8k [9] & high-school level MATH [14]. However, recent math specific competition and Olympiad-level benchmarking on Math Odyssey [11], American Invitational Mathematics Examination (AIME) & American Mathematics Competitions (AMC) [4, 10, 23] questions show that they continue to struggle with advanced mathematical reasoning.

**Related Work**: In recent times, numerous developments in multiple research directions have taken place to enhance the math ability of the LLMs. One of the major ones has been along the prompting and thinking strategies such as Chain-of-Thought (COT) method [31, 15] that has shown to evoke multi-step thinking in LLMs before arriving at the answer. These methods struggle with complex and symbolic computations. For this, PAL [12] & POT [7] suggest making LLMs perform reasoning by writing program and offloading the computations to code interpreter. Another line of research has been around pre-training and supervised fine-tuning (SFT). Multiple studies [24, 34, 10, 3, 16, 22, 25] have shown pre-training LLMs on maths tokens results in increased mathematical knowledge and reasoning abilities. Recent approaches [36, 13, 37, 28, 24, 27, 19, 4, 33, 26] have tried creating synthetic reasoning paths/trajectories using a teacher model like GPT4 [1] for SFT. Also, some studies [29, 35, 32, 6, 17] provide an alternative to manual annotations for process supervision [18].

---

[*]Corresponding author: Kunal Singh (kunal.singh@fractal.ai)

38th Conference on Neural Information Processing Systems (NeurIPS 2024) MATH-AI.

**Motivation**: COT prompting helps LLMs to solve a problem using a step-by-step thought process. PAL & POT introduced problem-solving via program generation where the answer is generated by executing the generated program. ToRA [13] & Mathcoder [28] introduced tool-integrated math problem solving format. There, model outputs natural language reasoning followed by program generation to solve the problem in a single turn/block and incorporates code-interpreter output for either summarizing the program output to get the final answer and terminate; or re-attempt the problem in the subsequent turn using the same format. For brevity, let's call ToRA's defined way of tool-integrated reasoning (TIR) strategy as TIR-ToRA.

Fundamentally, both PAL & TIR-ToRA generate a single program block to solve the problem. Additionally, TIR-ToRA framework allows the model to re-attempt the program generation in case of execution error. These approaches show improved performance over COT on elementary & high school level math problems. However, solving olympiad level math requires coming up with complex and creative solutions. Often, it is not feasible to solve a complex problem entirely using a single program block and as a result, these strategies fail to systematically address each detailed step of the problem-solving process. It tends to overlook specified constraints, edge cases or necessary simplifications, which are often encountered in Olympiad-level problems.

**Our Contribution**: Olympiad level math problem-solving can be viewed as solving/exploring an intermediate sub-task in depth; and discovering + solving the next critical sub-task dynamically basis the accumulated knowledge of previous sub-tasks explorations. To this end, we propose Step-by-Step Coding paradigm (SBSC) which is a multi-turn math reasoning framework that leverages existing programming [20] and in-context learning skills [5] of the current generation of LLMs, particularly Claude-3.5-Sonnet [2] & GPT-4o [21]. It uses program generation as the reasoning strategy to solve an intermediate sub-task unlike PAL & TIR-ToRA. In each turn, it leverages code-interpreter results and knowledge of previous sub-tasks solutions to define and programmatically solve the next sub-task. We investigate the performance of SBSC on last 11 years of AIME & AMC-12 questions. We also benchmark on Olympiad-subset of MathOdyssey dataset. We compare our method with existing reasoning strategies: COT, PAL, TIR-ToRA. We conduct ablations to understand the benefits of our approach such as sensitivity to exemplars, topic-wise analysis and measuring improvement in program refinement/debugging ability over TIR-ToRA due to the granular nature of SBSC process.

## 2  Method

SBSC is a multi-turn, program-generation based math reasoning framework where at each turn: the model generates an intermediate sub-task and corresponding program to solve that sub-task by leveraging the outputs of the previous turns. At the end of each turn, code interpreter is used to execute the program block to generate the solution for the intermediate sub-task. The intermediate sub-task depends on the results of the previous turns and the question. The code snippet for the $i^{th}$ sub-task directly incorporates the execution results of the previous code snippets by directly defining them as variables and symbols. This way SBSC makes LLMs generate sequence of targeted programs over multiple turns to solve complex math problems.

Our inference procedure is inspired by ToRA [13]. Solution chain is initialized with the Prompt $p$ containing method instructions followed by exemplars and the current question $q$. At each step, LLM $G$ first outputs a subtask $s_i$. If $s_i$ generation ends with stop-word "`###END OF CODE`", we extract the final answer. Else, it continues to generate program code $c_i$ ending with stop-word "``` `output`". We then pass $c_i$ to code interpreter and obtain the execution message or output $o_i \leftarrow E(c_i)$. The solution chain is updated by concatenating it with $s_i, c_i, o_i$ and loop continues till we get "`###END OF CODE`". $\oplus$ denoting concatenation, the sequential process can be generalised as:

$$s_i \oplus c_i \sim G(\cdot \mid p \oplus q \oplus (s_1 \oplus c_1 \oplus o_1) \oplus (s_2 \oplus c_2 \oplus o_2) \oplus ......(s_{i-1} \oplus c_{i-1} \oplus o_{i-1})) \quad (1)$$

Step-wise sequential approach of SBSC ensures that every part of the problem is addressed with exact precision, reducing the risk of errors that might arise from false assumptions or skipped steps. Having separate programs for each part of the solution also allows it to make necessary simplifications that would make the future subparts, and hence the whole problem, easier to solve allowing for a more granular and precise approach to problem-solving compared to existing methods. In case the code execution at any step results in an erroneous output, SBSC is better able to rectify that particular step. Fig 1a shows a visual sample SBSC response for an AIME question and Fig 1b shows TIR-ToRA

response for the same question. More detailed discussion on comparison is at Appendix A.1. In depth understanding of SBSC via multiple examples and comparisons at Appendix A.4.

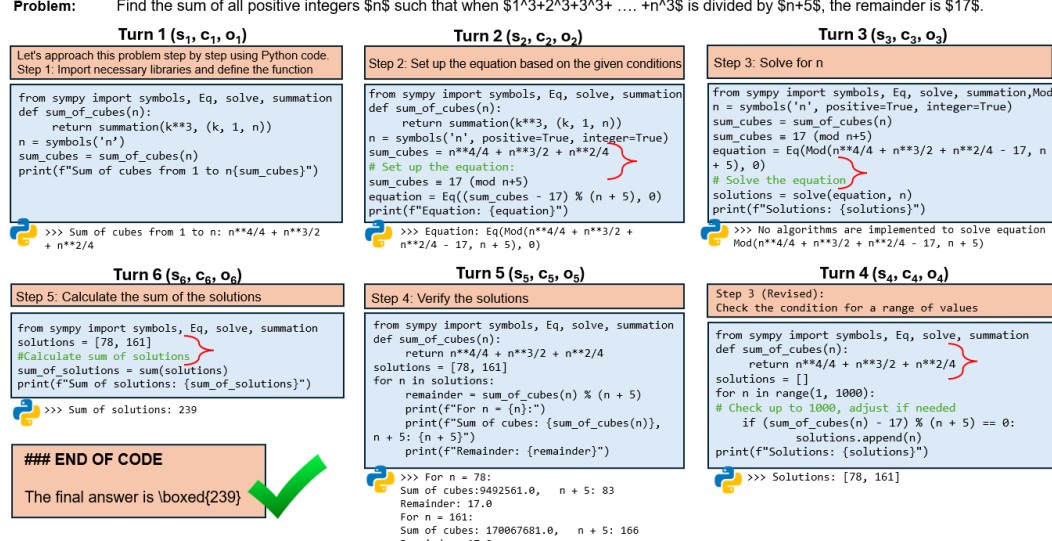

(a) Example multi-turn SBSC response for an AIME problem. Pink boxes denote the sub-task $s_i$ at the i-th step, blue boxes denote the program $c_i$ to solve $s_i$ and $>>>$ denote the corresponding execution output $o_i$. The red curly brackets indicate reusing outputs from earlier steps.

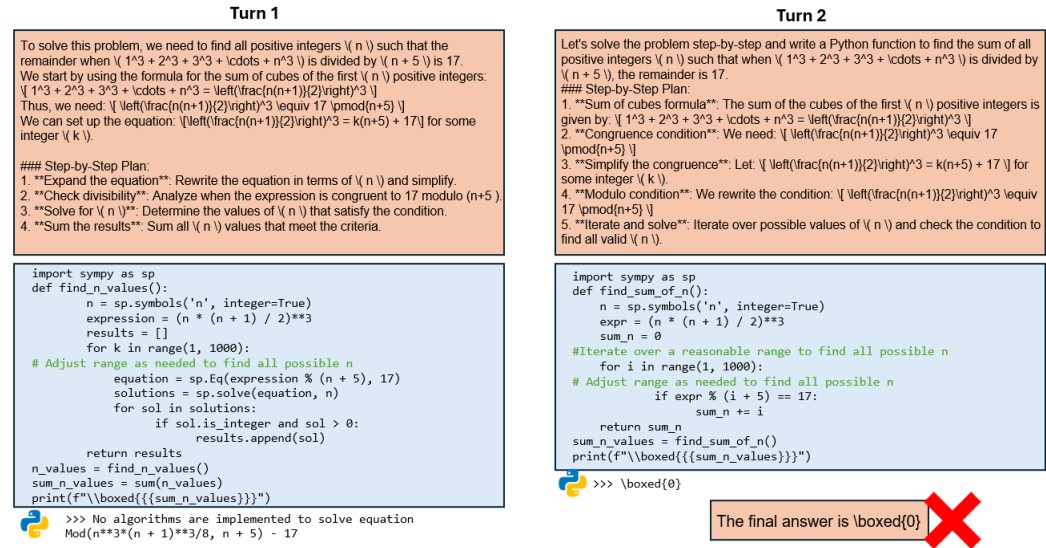

(b) Example TIR-ToRA response for the same problem, which is not solved correctly. In first turn, it tries to solves the problem at once using a rational and program. It encounters error and in second turn, tries to fix the entire approach and solve again but the solution is incorrect.

Figure 1: Comparison of SBSC and TIR-ToRA frameworks.

## 3 Experiment

**Benchmark datasets** We create our datasets using problems of last 11 years from popular math competitions AMC and AIME. We obtain questions and answers (Q&A) in LaTeX format from the AoPS Wiki website. We remove problems which are dependent on accompanying images and process the Q&A to have integer answers using GPT-4o if needed, leaving us with 330 AIME problems and 475 AMC-12 problems. We also use MathOdyssey [11], a popular benchmark for LLM math

reasoning, consisting of problems of varying difficulties. We include the 148 problems belonging to Olympiad-level competitions and perform similar filtering and processing. For more details on how we processed the dataset, please refer to Appendix A.2.

**Models & Configurations** We use `gpt-4o-2024-05-13` and `Claude-3.5-Sonnet` as base LLMs for our experiments. For all datasets and all reasoning frameworks, we use 4-shot setting. Maximum number of turns (n) for both TIR-ToRA and SBSC is set to 15. For greedy decoding inference, we use `temperature=0` and `max_tokens=1024` and also, we run 3 times and report average. Given SBSC is multi-turn in nature (on average 6-7 turns per problem , Table 2 in Appendix A.3) , we also benchmark SBSC's greedy decoding results against self-consistency (SC) [30] decoding results (majority@7) of COT, PAL & TIR-ToRA. For SC decoding, we use `temperature=0.7` and `top_p=0.9`. Note: we experimentally observe that for n > 4, there is no improvement in accuracy for TIR-ToRA so we set n=4 for TIR-ToRA during SC decoding. All ablations were conducted using `Claude-3.5-Sonnet` unless otherwise specified.

**Prompting/Few-shot Exemplars** For both AIME and AMC, we select 90 questions each, drawn from problems of years other than those included in the evaluation datasets. These questions were prompted with COT, PAL, TIR-ToRA and SBSC to generate corresponding solutions in accurate format. For each dataset, we create a subset of 10 problems correctly solved by every method and finally select a combination of 4 exemplars among them. For MathOdyssey, we use AIME exemplars as both are of similar difficulty level. We provide the 4 chosen exemplars and system-prompts, used in the main experiments, for different methods in Appendix (A.5, A.6, A.8, A.9) & repository here.

# 4 Results

**Main Results**

| Method | AMC | | AIME | | MathOdyssey | |
|---|---|---|---|---|---|---|
| | greedy | maj@7 | greedy | maj@7 | greedy | maj@7 |
| Claude-3.5-Sonnet | | | | | | |
| COT | 31.16 | 35.79 | 9.09 | 10.91 | 11.89 | 16.89 |
| PAL | 35.79 | 36.42 | $\underline{27.48}$ | $\underline{28.79}$ | 27.23 | 31.01 |
| TIR-ToRA | $\underline{38.59}$ | $\underline{43.16}$ | 24.64 | 26.67 | $\underline{27.23}$ | $\underline{32.43}$ |
| SBSC (Ours) | **49.33**$_{\uparrow 10.7}$ | $^{-}\uparrow 6.2$ | **35.45**$_{\uparrow 8}$ | $^{-}\uparrow 6.7$ | **39.86**$_{\uparrow 12.6}$ | $^{-}\uparrow 7.4$ |
| GPT-4o | | | | | | |
| COT | 35.94 | 37.47 | 10.39 | 12.12 | 13.51 | 17.57 |
| PAL | 36.48 | 38.11 | $\underline{24.63}$ | $\underline{26.97}$ | 15.74 | 20.27 |
| TIR-ToRA | $\underline{37.33}$ | $\underline{40.42}$ | 22.42 | 25.45 | $\underline{19.59}$ | $\underline{23.64}$ |
| SBSC (Ours) | **44.55**$_{\uparrow 7.2}$ | $^{-}\uparrow 4.1$ | **30.7**$_{\uparrow 6.1}$ | $^{-}\uparrow 3.7$ | **26.55**$_{\uparrow 7}$ | $^{-}\uparrow 2.9$ |

Table 1: Benchmarking SBSC against different math reasoning methods across 3 datasets:We report the average accuracy over 3 runs. Best result in each setting is highlighted in **bold** & second best is underlined. Absolute improvement in performance by SBSC over the previous best method in each setting is indicated in subscript.

As shown in Table 1, on AMC dataset, SBSC shows an absolute improvement over TIR-ToRA by roughly 11% using Claude-3.5-Sonnet and 7% using GPT-4o. SBSC greedy decoding results outperforms SC decoding results of TIR-TORA by absolute 6% and 4%, for Claude-3.5-Sonnet and GPT-4o respectively. We see similar absolute improvements in accuracy on our AIME dataset too. SBSC outperforms its nearest competitor (PAL) by 8% and 6% with greedy settings and SC settings by 6.7% and 3.7% for Claude-3.5-Sonnet and GPT-4o respectively. For MathOdyssey, SBSC improves by as much as 12.6% and 7% over TIR-ToRA while showing improvement of 7.4% and 3% over its SC variant, for Claude-3.5-Sonnet & GPT-4o respectively. Standard deviation values at A.10.

# 5 Ablations

**Sensitivity to Exemplars**: We study the effect of number/choice of examples in prompting on SBSC's performance. As shown in Figure 2, we observe a notable increase in performance when

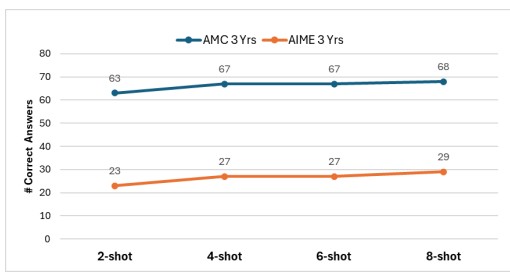

Figure 2: Effect of Number of Exemplars

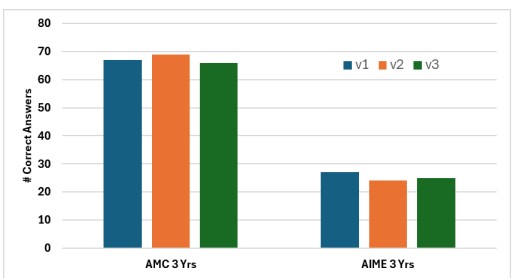

Figure 3: Sensitivity to choice of Exemplars

increasing the examples from 2 to 4, which then starts to saturate as we further increase the number of examples to 6 and 8. This justifies our decision of using a 4-shot setting. To understand if the choice of exemplars affect the accuracy or not, we conduct a sensitivity analysis. We randomly sample 4 exemplars out of the already created pool of 10 exemplars three times to create 3 variations of 4-shot prompts: v1, v2, and v3. In Figure 3, we can see that the performance remains stable irrespective of the exemplars used, across a subset of AIME (2022-2024) and AMC (2021-2023) problems.

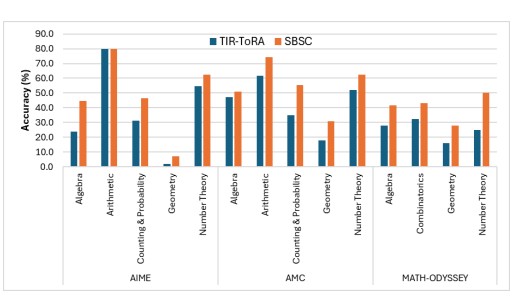

Figure 4: Topic breakdown analysis

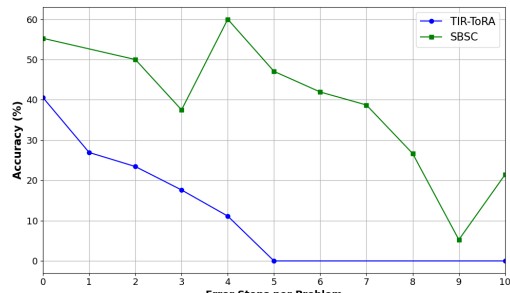

Figure 5: Comparison of Debugging Abilities

**Topic-wise Analysis**: We use GPT-4o-mini [21] to classify problems from AIME and AMC, while MathOdyssey already had topic labels. As can be seen in Figure 4, our method outperforms TIR-ToRA in all the individual topics and across all 3 datasets, thereby proving beneficial for all topics.

**Code Debugging Ability**: We present the superior ability of our method to resolve an error related to code execution. If at any step of the trajectory chain, the program returns an execution error, we consider that to be an error step. In Figure 5, we see that SBSC is able to recover from even multiple wrong steps and reach the correct final answer quite easily when compared to TIR-ToRA whose performance drops steeply on increasing error steps. This can be attributed to the fact that SBSC, being precise and granular, tackles only a specific part of the problem and finds it easier to correct its mistakes compared to TIR-ToRA which tries to correct the program at the problem level.

## 6 Conclusion

SBSC is a math reasoning framework that solves a problem by generating a sequence of sub-tasks and corresponding program blocks. Each sub-task and its corresponding program solution is generated leveraging the execution outputs and solutions of all the previous sub-tasks. We show performance improvements of SBSC over TIR-ToRA, PAL & COT on challenging math problems. **Limitations**: We only focus on text-based questions. We also just evaluate on integer-answer type questions.

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

## A   Appendix / supplemental material

Optionally include supplemental material (complete proofs, additional experiments and plots) in appendix. All such materials **SHOULD be included in the main submission.**

## A.1 Framework Explanation

We present example responses from both SBSC and TIR-ToRA for a problem from AIME in figures 1a and 1b respectively. As can be seen, in case of TIR-ToRA, the initial program generated by the model runs into an execution error. At the next turn, it attempts to rectify the error and comes up with a new approach and the corresponding program. This time, the code executes correctly but the final answer is wrong.

On the other hand, we see that SBSC is progressing step-by-step, tackling individual sub-tasks with separate programs and utilising outputs of previous steps. In the third step, it runs into a code execution error but succeeds in rectifying it using a different approach in the very next turn. Further, we observe SBSC checking the validity of the generated solutions in the fourth step before proceeding with the final step and ultimately reaches the correct answer.

This example also helps to illustrate how our approach is different from Least-to-Most (L2M) prompting [39] where the first stage involves pre-decomposing the question into two or more sub-questions in one go and then finding solutions for these pre-defined sub-questions whereas SBSC identifies sub-tasks on the fly, based on preceding steps' results and the final goal of the problem. It also doesnt use tool-integration. Major advantage of SBSC is the granular program/sub-task level thinking/refinement ability which previous works lack.

## A.2 Dataset Processing

All AIME problems have a unique integer answer ranging from 0 to 999, while AMC-12 problems are of Multiple Choice Question(MCQ) format. Following Numina AIMO, we remove all the answer choices from each AMC-12 question and modify the question, wherever necessary, to ensure an integer answer. For this, we prompt GPT-4o to append an additional line at the end of each problem as suitable. Following is an example for demonstration:

**Original Question:** An urn contains one red ball and one blue ball. A box of extra red and blue balls lies nearby. George performs the following operation four times: he draws a ball from the urn at random and then takes a ball of the same color from the box and returns those two matching balls to the urn. After the four iterations the urn contains six balls. What is the probability that the urn contains three balls of each color?
Answer: $\frac{1}{5}$

**Modified Question:** An urn contains one red ball and one blue ball. A box of extra red and blue balls lies nearby. George performs the following operation four times: he draws a ball from the urn at random and then takes a ball of the same color from the box and returns those two matching balls to the urn. After the four iterations the urn contains six balls. What is the probability that the urn contains three balls of each color? If the answer is represented as a fraction $\frac{m}{n}$ in its simplest terms, what is the value of m+n?
Integer Answer: 6

## A.3 Number of Steps in SBSC

In Table 2, we present the number of turns taken per question by SBSC across the different datasets.

## A.4 Understanding SBSC in Detail

In this section, we demonstrate some scenarios where SBSC has been successful while TIR-ToRA has failed, with the help of some example questions and investigating the responses obtained from the two models.

Let's consider the question in Example 1, involving a geometric progression of numbers written in logarithmic form, which TIR-ToRA gets wrong.The method uses a binary search technique, which is not very precise when dealing with exact values required for mathematical problems, especially when fractions are involved.The solution uses a function to check whether the logarithms form a geometric progression which introduces additional complexity and potential inaccuracies because it involves comparing ratios that may not be exactly equal due to floating-point arithmetic.Also, this single-turn method tends to overlook specified constraints or necessary simplifications, which are often encountered in Olympiad level problems and instead makes false assumptions.

| Number of turns or steps | AMC | AIME | MathOdyssey |
|---|---|---|---|
| 2 | 21 | 12 | 8 |
| 3 | 57 | 19 | 17 |
| 4 | 101 | 47 | 19 |
| 5 | 79 | 51 | 21 |
| 6 | 63 | 43 | 28 |
| 7 | 41 | 43 | 14 |
| 8 | 42 | 31 | 10 |
| 9 | 12 | 18 | 8 |
| others | 59 | 66 | 23 |
| **Average turns or steps/Problem** | 6.0 | 6.9 | 6.4 |

Table 2: Table showing number of turns/steps used by SBSC

The question in Example 2 is an example scenario where TIR-ToRA fails because it makes an incorrect assumption. It misinterprets the Lipschitz condition and incorrectly makes a simpler assumption that the difference $f(800) - f(400)$ is equal to the maximum possible difference, which is 200. While the magnitude of the difference is bounded by 200, it does not mean that the actual difference will always be 200. Iterative solutions, as are often the only way out in single program based solutions, can sometimes lead to infinite loops, especially in cases where the stopping condition is not clearly defined or understood by the LLM.

As can be seen in Example 3, the single code is unable to take advantage of the factorization of $20^{20}$, which is key to solving the problem efficiently and instead iterates over a very large range of potential values for $m$, leading to inefficiency. The upper bound 2020 is extremely large and the sheer number of iterations causes a timeout.

Example 4 presents a scenario where TIR-ToRA makes up an assumption about the problem and writes the code for terminating a loop accordingly, which leads to a timeout error, as the incorrect assumption leads to an infinite loop. It lacks intermediate checks that would provide insights into whether the sequence terms are of the form $\frac{t}{t+1}$, which is crucial for solving the problem and would have enabled it to chalk out the termination conditions suitably.

On the other hand, our Step-By-Step Coding method enforces a decomposition of the problem into smaller sub-task. Each sub-task is tackled independently by the LLM, which generates code to solve it and then uses the resulting output to suitably proceed to the next sub-task and this process continues till the final answer is reached. Such an approach ensures that every part of the problem is addressed with exact precision, reducing the risk of errors that might arise from skipped steps. Dividing the problem into multiple sub-tasks also allows it to make necessary simplifications that would make the future sub-tasks, and hence the entire problem, easier to solve.

Going back to the problem in Example 1, SBSC starts by defining the logarithms and setting up the equations based on the geometric progression condition. It then simplifies the equations to reduce them to a more manageable form, eliminating unnecessary complexity and allowing straightforward solving. Throughout the problem, it uses precise mathematical formulations of the problem, ensuring the solution is accurate. Since this method isn't trying to solve the entire problem at one go, it doesn't need to make any assumptions to simplify the problem statement.

For the question in Example 2, it correctly interprets the problem, keeps applying the given Lipschitz condition as it solves each sub-task and finds the correct maximum possible value of $f(f(800)) - f(f(400))$. By systematically checking for constraints and edge cases at each stage, our method guarantees that solutions are not only accurate but also valid under all possible conditions. This is particularly crucial for problems with intricate conditions or multiple solution paths. Additionally, our process explicitly defines loop conditions to ensure that termination criteria are clear, allowing for correction if issues like infinite loops arise.

For the problem in Example 3, the first sub-task correctly identified by SBSC is to factorize the large number $20^{20}$ into smaller powers, paving the path towards an efficient solution. In the subsequent steps, it takes care of constraints like ensuring that m and n are integers and avoids over-counting by properly iterating over valid powers, giving us the correct answer.

Likewise, in Example 4, we see that our method generates each term of the sequence one at a time, allowing for intermediate checks to confirm whether each term is of the desired form $\frac{t}{t+1}$. Thus, it accurately identifies the terms that meet the required condition and ensures that only the correct

indices are considered. This avoids the infinite loop risk and ensures the correctness of each step before moving on. We also notice that the model runs into an error at the first sub-task itself but based on the corresponding code output, it is able to take an alternate approach and successfully solves the sub-task, moving onto later ones. This example also highlights our superior code debugging ability, as discussed earlier.

**Example Question 1:** There is a unique positive real number x such that the three numbers $\log_8 2x$, $\log_4 x$, and $\log_2 x$, in that order, form a geometric progression with positive common ratio. The number x can be written as $\frac{m}{n}$, where m and n are relatively prime positive integers. Find m + n

---

**Incorrect solution by TIR-ToRA**

1 Let's approach this problem step-by-step:

2     1. Define a function to check if three numbers form a geometric progression.

3     2. Use binary search to find $x$ that satisfies the condition.

4     3. Convert $x$ to a fraction and return the sum of its numerator and denominator.

```python
5  import math
6  from fractions import Fraction
7  def solution():
8      # Function to check if three numbers form
9      #a geometric progression
10     def is_geometric_progression(a, b, c):
11         ratio1 = b / a
12         ratio2 = c / b
13         return math.isclose(ratio1, ratio2) and ratio1 > 1
14     # Binary search to find x
15     left, right = 1, 1000
16     while right - left > 1e-10:
17         x = (left + right) / 2
18         log8_2x = math.log(2*x, 8)
19         log4_x = math.log(x, 4)
20         log2_x = math.log(x, 2)
21         if is_geometric_progression(log8_2x, log4_x, log2_x):
22             break
23         elif (log4_x / log8_2x) > (log2_x / log4_x):
24             left = x
25         else:
26             right = x
27     # Convert x to a fraction
28     frac = Fraction(x).limit_denominator()
29     return frac.numerator + frac.denominator
30 result = solution()
31 print(f"\\boxed{{{result}}}")
32
33 >>> output:
34 \boxed{2}
```

35 The answer is $\boxed{2}$

---

**Correct solution by SBSC:**

36 Let's solve this problem step by step using SymPy-based Python code.
37 Step 1: Define the symbols and set up the equations

```python
38 from sympy import symbols, log, Eq, solve
39 # Define symbols
40 x, r = symbols('x r')
41 # Define the three logarithms
42 log1 = log(2*x, 8)
43 log2 = log(x, 4)
```

```
44  log3 = log(x, 2)
45  # Set up equations for geometric progression
46  eq1 = Eq(log2 / log1, r)
47  eq2 = Eq(log3 / log2, r)
48  print("Equation 1:", eq1)
49  print("Equation 2:", eq2)
50
51  >>> output
52  Equation 1: Eq(log(8)*log(x)/(log(4)*log(2*x)), r)
53  Equation 2: Eq(log(4)/log(2), r)
```

Let's continue solving this problem step by step using SymPy-based Python code.
Step 2: Simplify the equations

```
56  from sympy import symbols, log, Eq, solve, simplify
57  # Define symbols
58  x, r = symbols('x r')
59  # Define the three logarithms
60  log1 = log(2*x, 8)
61  log2 = log(x, 4)
62  log3 = log(x, 2)
63  # Set up equations for geometric progression
64  eq1 = Eq(log2 / log1, r)
65  eq2 = Eq(log3 / log2, r)
66  # Simplify equations
67  eq1_simplified = simplify(eq1)
68  eq2_simplified = simplify(eq2)
69  print("Simplified Equation 1:", eq1_simplified)
70  print("Simplified Equation 2:", eq2_simplified)
71
72  >>> output:
73  Simplified Equation 1: Eq(r, 3*log(x)/(2*log(x) + 2*log(2)))
74  Simplified Equation 2: Eq(r, 2)
```

Let's continue solving this problem step by step using SymPy-based Python code.
Step 3: Solve the equations

```
77  from sympy import symbols, log, Eq, solve, simplify
78  # Define symbols
79  x, r = symbols('x r')
80  # Define the simplified equations
81  eq1 = Eq(r, 3*log(x)/(2*log(x) + 2*log(2)))
82  eq2 = Eq(r, 2)
83  # Solve the system of equations
84  solution = solve((eq1, eq2), (x, r))
85  print("Solution:", solution)
86
87  >>> output
88  Solution: [(1/16, 2)]
```

Let's continue solving this problem step by step using SymPy-based Python code.
Step 4: Extract the value of x and convert to a fraction

```
91   from sympy import symbols,log,Eq, solve, simplify, Rational
92   # Define symbols
93   x, r = symbols('x r')
94   # Define the solution from the previous step
95   solution = [(Rational(1, 16), 2)]
96   # Extract the value of x
97   x_value = solution[0][0]
98   print("x =", x_value)
99   # Convert x to a fraction
100  m = x_value.numerator
101  n = x_value.denominator
102  print("x as a fraction: {}/{}".format(m, n))
```

```
103
104  >>> output
105  x = 1/16
106  x as a fraction: 1/16
```

Let's continue solving this problem step by step using SymPy-based Python code.
Step 5: Calculate m + n

```
109  from sympy import symbols,log,Eq, solve, simplify, Rational
110  # Define symbols
111  x, r = symbols('x r')
112  # Define the solution from the previous step
113  solution = [(Rational(1, 16), 2)]
114  # Extract the value of x
115  x_value = solution[0][0]
116  # Convert x to a fraction
117  m = x_value.numerator
118  n = x_value.denominator
119  # Calculate m + n
120  result = m + n
121  print("m =", m)
122  print("n =", n)
123  print("m + n =", result)
124
125  >>> output:
126  m = 1
127  n = 16
128  m + n = 17
```

### END OF CODE
The final answer is $\boxed{17}$

**Example Question 2:** Consider functions $f$ that satisfy $|f(x) - f(y)| \leq 0.5|x - y|$ for all real numbers $x$ and $y$. Of all such functions that also satisfy the equation $f(300) = f(900)$, what is the greatest possible value of $f(f(800) - f(f(400)))$?

---

**Incorrect solution by TIR-ToRA:**

To solve this problem, we need to consider the properties of the given function and find the maximum possible difference between $f(f(800))$ and $f(f(400))$. Let's solve it using python:

```
133  def solution():
134      #Given: |f(x) - f(y)| <= 0.5|x - y| for all real x and y
135      #Also, f(300) = f(900)
136      #Maximum difference between f(800) and f(400)
137      max_diff_800_400 = 0.5 * (800 - 400)   # 200
138      #Maximum value of f(800) - f(400)
139      max_f800_f400 = max_diff_800_400   # 200
140      #Maximum difference between f(f(800)) and f(f(400))
141      max_final_diff = 0.5 * max_f800_f400   # 100
142      return max_final_diff
143  result = solution()
144  print(f"\\boxed{{{result}}}")
145
146  >>> output:
147  \boxed{100}
```

The answer is $\boxed{100}$

Let's approach this problem step by step using SymPy-based Python code where appropriate.
Step 1: Understand the given condition The condition |f(x) - f(y)| 1/2|x - y| implies that f is a Lipschitz continuous function with Lipschitz constant 1/2. This means f doesn't change too rapidly.
Step 2: Analyze the equation f(300)=f(900).This equation tells us that f takes the same value at x=300 and x=900. Let's call this value k.

```python
from sympy import symbols, Abs
k = symbols('k')
print(f"f(300) = f(900) = {k}")

>>> output:
f(300) = f(900) = k
```

Let's continue solving this problem step by step.
Step 3: Apply the Lipschitz condition to f(800) and f(400) We can use the Lipschitz condition to find bounds for f(800) and f(400) in terms of k.

```python
from sympy import symbols, Abs, Min, Max
k = symbols('k')
# For f(800)
f_800_min = k - Abs(800 - 900)/2
f_800_max = k + Abs(800 - 900)/2
# For f(400)
f_400_min = k - Abs(400 - 300)/2
f_400_max = k + Abs(400 - 300)/2
print(f"f(800) is between {f_800_min} and {f_800_max}")
print(f"f(400) is between {f_400_min} and {f_400_max}")

>>> output:
f(800) is between k - 50 and k + 50
f(400) is between k - 50 and k + 50
```

Let's continue solving this problem step by step.
Step 4: Analyze $f(f(800))$ and $f(f(400))$ Now we need to apply the Lipschitz condition again to $f(f(800))$ and $f(f(400))$.

```python
from sympy import symbols, Abs, Min, Max
k = symbols('k')
# For f(f(800))
ff_800_min = k - Abs(k + 50 - k)/2
ff_800_max = k + Abs(k - 50 - k)/2
# For f(f(400))
ff_400_min = k - Abs(k + 50 - k)/2
ff_400_max = k + Abs(k - 50 - k)/2
print(f"f(f(800)) is between {ff_800_min} and {ff_800_max}")
print(f"f(f(400)) is between {ff_400_min} and {ff_400_max}")

>>> output
f(f(800)) is between k - 25 and k + 25
f(f(400)) is between k - 25 and k + 25
```

Let's continue solving this problem step by step.
Step 5: Calculate the maximum possible difference Now we need to find the maximum possible value of $f(f(800)) - f(f(400))$.

```python
from sympy import symbols, Abs, Min, Max
k = symbols('k')
# Maximum possible difference
max_diff = (k + 25) - (k - 25)
print(f"The maximum possible difference f(f(800))-f(f(400))
is {max_diff}")
```

```
205  >>> output:
206  The maximum possible difference f(f(800)) - f(f(400)) is 50
```

```
207  ### END OF CODE
208  The final answer is  50
```

**Example Question 3:** Find the number of ordered pairs of positive integers $(m, n)$ such that $m^2 n = 20^{20}$

---

**Incorrect solution by TIR-ToRA:**

209  Let's approach this problem step-by-step:

210      1. Iterate through possible values of m up to the square root of $20\hat{2}0$

211      2. For each m, check if $m^2$ divides $20^{20}$ evenly

212      3. Return the total count of valid pairs

```python
213  import math
214  def solution():
215      count = 0
216      target = 20**20
217      for m in range(1, int(math.sqrt(target)) + 1):
218          if target % (m**2) == 0:
219              n = target // (m**2)
220              count += 1
221      return count
222  result = solution()
223  print(f"The number of ordered pairs (m,n) is:
224  f"\\boxed{{{result}}}"")
225
226  >>> output:
227  TimeoutError: [Errno 110] Connection timed out
```

228  Looks like we got a timeout error, let's try a different approach that should not result in any
229  error:

```python
230  import math
231  def solution():
232      count = 0
233      target = 20**20
234      for m in range(1, int(math.sqrt(target)) + 1):
235          if target % (m**2) == 0:
236              n = target // (m**2)
237              count += 1
238      return count
239  result = solution()
240  print(f"The number of ordered pairs (m,n) is:
241  f"\\boxed{{{result}}}"")
242
243  >>> output:
244  TimeoutError: [Errno 110] Connection timed out
```

245  ...
246  This loop continues without ultimately getting an answer.

---

**Correct solution by SBSC:**

247  Let's approach this problem step by step using SymPy-based Python code.
248  Step 1: Define the equation and initialize variables

```python
249  from sympy import symbols,Eq,solve, divisors, prime_factors
250  m, n = symbols('m n', positive=True, integer=True)
```

```
251  equation = Eq(m**2 * n, 20**20)
252  print("Equation:", equation)
253
254  >>> output:
255  ImportError: cannot import name 'prime_factors' from 'sympy'
```

I apologize for the error. It seems the 'prime factors' function is not available in the version of SymPy you're using. Let's modify our approach and use the 'factorint' function instead, which is more commonly available. Here's the corrected code:

Step 1: Define the equation and initialize variables

```
260  from sympy import symbols, Eq, solve, divisors, factorint
261  m, n = symbols('m n', positive=True, integer=True)
262  equation = Eq(m**2 * n, 20**20)
263  print("Equation:", equation)
264
265  >>> output
266  Equation: Eq(m**2*n, 104857600000000000000000000)
```

Let's continue solving this problem step by step.

Step 2: Factor $20^{20}$ and analyze its structure

```
269  from sympy import symbols, Eq, solve, divisors, factorint
270  m, n = symbols('m n', positive=True, integer=True)
271  equation = Eq(m**2 * n, 20**20)
272  # Factor 20^20
273  factorization = factorint(20**20)
274  print("Factorization of 20^20:", factorization)
275  # Analyze the structure
276  power_of_2 = factorization[2]
277  power_of_5 = factorization[5]
278  print("Power of 2:", power_of_2)
279  print("Power of 5:", power_of_5)
280
281  >>> output
282  Factorization of 20^20: {2: 40, 5: 20}
283  Power of 2: 40
284  Power of 5: 20
```

Let's continue solving this problem step by step.
Step 3: Determine possible values for m and n

```
287  from sympy import symbols, Eq, solve, divisors, factorint
288  m, n = symbols('m n', positive=True, integer=True)
289  equation = Eq(m**2 * n, 20**20)
290  # Factor 20^20
291  factorization = factorint(20**20)
292  power_of_2 = factorization[2]
293  power_of_5 = factorization[5]
294  # Initialize counter for valid pairs
295  valid_pairs = 0
296  # Iterate through possible values of m
297  for m_power_of_2 in range(0, power_of_2 + 1, 2):
298      for m_power_of_5 in range(0, power_of_5 + 1):
299          # Calculate m
300          m_value = 2**m_power_of_2 * 5**m_power_of_5
301          # Calculate corresponding n
302          n_power_of_2 = power_of_2 - 2*m_power_of_2
303          n_power_of_5 = power_of_5 - m_power_of_5
304          # Check if n is a positive integer
305          if n_power_of_2 >= 0 and n_power_of_5 >= 0:
306              valid_pairs += 1
307  print("Number of valid (m,n) pairs:", valid_pairs)
308
309  >>> output
```

```
310 | Number of valid (m,n) pairs: 231
```

311 | ### END OF CODE
312 | The final answer is $\boxed{231}$

**Example Question 4:** Consider the sequence $(a_k)\_k \geq 1$ of positive rational numbers defined by $a_1 = \frac{2020}{2021}$ and for $k \geq 1$, if $a_k = \frac{m}{n}$ for relatively prime positive integers m and n, then $a_{k+1} = \frac{m+18}{n+19}$. Determine the sum of all positive integers j such that the rational number $a_j$ can be written in the form $\frac{t}{t+1}$ for some positive integer t

### Incorrect solution by TIR-ToRA:

313 | Let's proceed step-by-step:

314 | 1. Generate the sequence of fractions starting from $a_1 = \frac{2020}{2021}$.

315 | 2. For each term, check if it is in the form $\frac{t}{t+1}$ for some integer $t$. If so, add its index to a
316 | running sum.

317 | 3. Generate the next term using the given formula.

318 | 4. Continue this process until we cycle back to the initial fraction. The sum of indices
319 | where the fraction is in the form $\frac{t}{t+1}$ is our final answer.

```python
320 | from fractions import Fraction
321 | def next_term(a):
322 |     return Fraction(a.numerator + 18, a.denominator + 19)
323 | def is_of_form_t_over_t_plus_1(frac):
324 |     return frac.denominator == frac.numerator + 1
325 | def solution():
326 |     a = Fraction(2020, 2021)
327 |     j_sum = 0
328 |     j = 1
329 |     while True:
330 |         if is_of_form_t_over_t_plus_1(a):
331 |             j_sum += j
332 |         a = next_term(a)
333 |         j += 1
334 |         # Check if we've reached the initial fraction again
335 |         if a == Fraction(2020, 2021):
336 |             break
337 |     return j_sum
338 | result = solution()
339 | print(f"\\boxed{{{result}}}")
```

Similar to the earlier example, this code results in a TimeOut error and then the loop keeps repeating.

### Correct solution by SBSC:

340 | Let's solve this problem step by step using SymPy-based Python code.
341 | Step 1: Define the sequence and initial value

```python
342 | from sympy import Rational, gcd
343 | def next_term(a):
344 |     m, n = a.numerator, a.denominator
345 |     return Rational(m + 18, n + 19)
346 | a = [Rational(2020, 2021)]
347 | print(f"a[1] = {a[0]}")
348 |
349 | >>> output
350 | a[1] = 2020/2021
```

Let's continue solving this problem step by step using SymPy-based Python code.
Step 2: Generate the sequence until we find a term of the form t/(t+1)

```
from sympy import Rational, gcd
def next_term(a):
    m, n = a.numerator, a.denominator
    return Rational(m + 18, n + 19)
a = [Rational(2020, 2021)]
j = 1
result = []
while True:
    if a[-1].denominator == a[-1].numerator + 1:
        result.append(j)
        print(f"Found a[{j}] = {a[-1]} of the form t/(t+1)")
    j += 1
    a.append(next_term(a[-1]))
    if j > 100:  # Limit to prevent infinite loop
        break
print(f"Indices found: {result}")

>>> output
Found a[1] = 2020/2021 of the form t/(t+1)
Found a[2] = 1019/1020 of the form t/(t+1)
Found a[8] = 161/162 of the form t/(t+1)
Found a[18] = 31/32 of the form t/(t+1)
Found a[30] = 19/20 of the form t/(t+1)
Indices found: [1, 2, 8, 18, 30]
```

Let's continue solving this problem step by step using SymPy-based Python code.
Step 3: Calculate the sum of the indicesfrom sympy import Rational, gcd

```
def next_term(a):
    m, n = a.numerator, a.denominator
    return Rational(m + 18, n + 19)
a = [Rational(2020, 2021)]
j = 1
result = []
while True:
    if a[-1].denominator == a[-1].numerator + 1:
        result.append(j)
    j += 1
    a.append(next_term(a[-1]))
    if j > 100:  # Limit to prevent infinite loop
        break
sum_of_indices = sum(result)
print(f"Indices found: {result}")
print(f"Sum of indices: {sum_of_indices}")

>>> output
Indices found: [1, 2, 8, 18, 30]
Sum of indices: 59
```

### END OF CODE
The final answer is $\boxed{59}$

### A.5 PAL Exemplars

In this section, we provide the prompts for Program-Aided Language models (PAL) method. We initially used the default prompt as mentioned in the original PAL paper, but the results were poor. We noticed that the response often contained textual reasoning before or after the program, which isn't the desired format for PAL. Hence, we modify the instructions to confine the responses only to include Python program and subsequently, also notice improved accuracy.

**For AIME**

Let's use python program to solve math problems.
DO NOT USE ANY TEXTUAL REASONING.
Your response must start with: "'python
Your response must end with: print(result)

Here are some examples you may refer to.

**Example Problem:** A frog begins at $P_0 = (0,0)$ and makes a sequence of jumps according to the following rule: from $P_n = (x_n, y_n)$, the frog jumps to $P_{n+1}$, which may be any of the points $(x_n + 7, y_n + 2)$, $(x_n + 2, y_n + 7)$, $(x_n - 5, y_n - 10)$, or $(x_n - 10, y_n - 5)$. There are $M$ points $(x, y)$ with $|x| + |y| \leq 100$ that can be reached by a sequence of such jumps. Find the remainder when $M$ is divided by 1000.

**Example Solution:**

```python
def solution():
    jumps = [(7, 2), (2, 7), (-5, -10), (-10, -5)]
    # Set to keep track of all reachable points, starting from the origin
     (0, 0).
    reachable = set([(0, 0)])
    # Queue to process points, starting with the origin (0, 0).
    queue = [(0, 0)]
    # Breadth-first search (BFS) to explore reachable points.
    while queue:
        # Pop the first point from the queue.
        x, y = queue.pop(0)
        # Iterate over all possible jumps.
        for dx, dy in jumps:
            # Calculate new coordinates after the jump.
            nx, ny = x + dx, y + dy
            # Check if the Manhattan distance is within 100 and the point
    hasn't been visited.
            if abs(nx) + abs(ny) <= 100 and (nx, ny) not in reachable:
                # Add the new point to the reachable set.
                reachable.add((nx, ny))
                # Add the new point to the queue to explore further.
                queue.append((nx, ny))
    return len(reachable) % 1000
result = solution()
print(result)
```

**Example Problem:** The AIME Triathlon consists of a half-mile swim, a 30-mile bicycle ride, and an eight-mile run. Tom swims, bicycles, and runs at constant rates. He runs fives times as fast as he swims, and he bicycles twice as fast as he runs. Tom completes the AIME Triathlon in four and a quarter hours. How many minutes does he spend bicycling?

**Example Solution:**

```python
from sympy import symbols, Eq, solve, Rational
def solution():
    x = symbols('x')
    # Set up the equation
    eq = Eq(Rational(1,2)/x + 30/(10*x) + 8/(5*x), Rational(17,4))
    # Solve the equation
    solution = solve(eq)[0]
    # Calculate bicycling time in hours
    bike_time = 30 / (10 * solution)
    # Convert to minutes
    bike_time_minutes = int(bike_time * 60)
    return bike_time_minutes
result = solution()
print result
```

**Example Problem:** Let $S$ be the increasing sequence of positive integers whose binary representation has exactly 8 ones. Let $N$ be the 1000th number in $S$. Find the remainder when $N$ is divided by 1000

**Example Solution:**

```python
def solution():
    count = 0  # Initialize a counter to track how many numbers have been
    found
    n = 1  # Start checking numbers from 1 upwards
    while count < 1000:  # Continue the loop until we find the 1000th
    number
        # Check if the binary representation of the number 'n' has
    exactly 8 '1's
        if bin(n).count('1') == 8:
            count += 1  # Increment the counter when a number with 8 '1's
    is found
            # If this is the 1000th such number, return the remainder of
    n divided by 1000
            if count == 1000:
                return n % 1000
        n += 1  # Move to the next number
result = solution()
print(result)
```

**Example Problem:** Two geometric sequences $a_1, a_2, a_3, \ldots$ and $b_1, b_2, b_3, \ldots$ have the same common ratio, with $a_1 = 27\ b_1 = 99$, and $a_{15} = b_{11}$. Find $a_9$

**Example Solution:**

```python
def solution():
    # Initialize known values
    a1 = 27
    b1 = 99
    # Calculate the common ratio
    # We know that a15 = b11, so:
    # a1 * r^14 = b1 * r^10
    # 27 * r^14 = 99 * r^10
    # 27 * r^4 = 99
    # r^4 = 99/27 = 11/3
    r = (11/3) ** (1/4)
    # Calculate a9
    a9 = a1 * (r ** 8)
    return round(a9)
result = solution()
print(result)
```

**For AMC:**
Let's use python program to solve math problems.
DO NOT USE ANY TEXTUAL REASONING.
Your response must start with: "'python
Your response must end with: print(result)

Here are some examples you may refer to.

**Example Problem:** Small lights are hung on a string 6 inches apart in the order red, red, green, green, green, red, red, green, green, green, and so on continuing this pattern of 2 red lights followed by 3 green lights. How many feet separate the 3rd red light and the 21st red light? Note: 1 foot is equal to 12 inches.

**Example Solution:**

```python
def solution():
    # Find position of 3rd red light
    n_3rd = 3
    complete_cycles_3rd = (n_3rd - 1) // 2
    remaining_lights_3rd = (n_3rd - 1) % 2
```

```
        pos_3rd = complete_cycles_3rd * 5 * 6 + remaining_lights_3rd * 6
        # Find position of 21st red light
        n_21st = 21
        complete_cycles_21st = (n_21st - 1) // 2
        remaining_lights_21st = (n_21st - 1) % 2
        pos_21st = complete_cycles_21st * 5 * 6 + remaining_lights_21st * 6
        # Calculate the distance in inches
        distance_inches = pos_21st - pos_3rd
        # Convert to feet
        distance_feet = distance_inches / 12
        return distance_feet
result = solution()
print(result)
```

**Example Problem:** A fruit salad consists of blueberries, raspberries, grapes, and cherries. The fruit salad has a total of 280 pieces of fruit. There are twice as many raspberries as blueberries, three times as many grapes as cherries, and four times as many cherries as raspberries. How many cherries are there in the fruit salad?

**Example Solution:**

```
from sympy import symbols, Eq, solve
def solution():
    # Define the symbols for the variables
    b, r, g, c = symbols('b r g c')
    # Define the equations based on the problem statement
    eq1 = Eq(r, 2*b)            # Equation 1: r = 2b
    eq2 = Eq(g, 3*c)            # Equation 2: g = 3c
    eq3 = Eq(c, 4*r)            # Equation 3: c = 4r
    eq4 = Eq(b + r + g + c, 280)  # Equation 4: b + r + g + c = 280
    # Solve the system of equations
    sol = solve((eq1, eq2, eq3, eq4))
    return sol[c]
result = solution()
print(result)
```

**Example Problem:** Last summer $30\%$ of the birds living on Town Lake were geese, $25\%$ were swans, $10\%$ were herons, and $35\%$ were ducks. What percent of the birds that were not swans were geese?

**Example Solution:**

```
def solution():
    # Total percentage of all birds
    total = 100
    # Percentages of each bird type
    geese = 30
    swans = 25
    herons = 10
    ducks = 35
    # Calculate percentage of birds that are not swans
    not_swans = total - swans
    # Calculate percentage of geese among birds that are not swans
    geese_among_not_swans = (geese / not_swans) * 100
    # Round to nearest whole number
    return round(geese_among_not_swans)
result = solution()
print(result)
```

Example Problem: At a twins and triplets convention, there were 9 sets of twins and 6 sets of triplets, all from different families. Each twin shook hands with all the twins except his/her siblings and with half the triplets. Each triplet shook hands with all the triplets except his/her siblings and with half the twins. How many handshakes took place?

**Example Solution:**

```python
def solution():
    # Number of twins and triplets
    twins = 9 * 2
    triplets = 6 * 3
    # Handshakes between twins
    twin_handshakes = (twins * (twins - 2)) // 2
    # Handshakes between triplets
    triplet_handshakes = (triplets * (triplets - 3)) // 2
    # Handshakes between twins and triplets
    twin_triplet_handshakes = (twins * triplets) // 2
    # Total handshakes
    total_handshakes = twin_handshakes + triplet_handshakes +
    twin_triplet_handshakes
    return total_handshakes
result = solution()
print(result)
```

## A.6  TIR-ToRA Exemplars

**For AIME**
Integrate step-by-step reasoning and Python code to solve math problems using the following guidelines:

- Analyze the question and write functions to solve the problem; the function should not take any arguments.
- Present the final result in LaTeX using a '□' without any units.
- Utilize the 'pi' symbol and 'Rational" from Sympy for $\pi$ and fractions, and simplify all fractions and square roots without converting them to decimal values.

Once you get the code output, just display the output answer within '□' and terminate.

Here are some examples you may refer to:

**Example Problem:** A frog begins at $P_0 = (0,0)$ and makes a sequence of jumps according to the following rule: from $P_n = (x_n, y_n)$, the frog jumps to $P_{n+1}$, which may be any of the points $(x_n + 7, y_n + 2)$, $(x_n + 2, y_n + 7)$, $(x_n - 5, y_n - 10)$, or $(x_n - 10, y_n - 5)$. There are $M$ points $(x, y)$ with $|x| + |y| \leq 100$ that can be reached by a sequence of such jumps. Find the remainder when $M$ is divided by 1000.

**Example Solution:**
Let's approach this problem step-by-step:

1. We need to find all points (x, y) that the frog can reach within the given constraint |x| + |y| 100.

2. We'll use a breadth-first search (BFS) algorithm to explore all possible points.

3. We'll keep track of visited points to avoid revisiting them.

4. We'll count the number of unique points visited.

Here's the Python code to solve this:
```python
from collections import deque
def count_reachable_points():
    # Possible jumps
    jumps = [(7, 2), (2, 7), (-5, -10), (-10, -5)]
    # Set to store visited points
    visited = set([(0, 0)])
    # Queue for BFS
    queue = deque([(0, 0)])
    while queue:
        x, y = queue.popleft()
        for dx, dy in jumps:
            new_x, new_y = x + dx, y + dy
            # Check if the new point is within the constraint and not
    visited
```

```
            if abs(new_x) + abs(new_y) <= 100 and (new_x, new_y) not in
    visited:
                visited.add((new_x, new_y))
                queue.append((new_x, new_y))
    # Count of reachable points is the size of visited set
    M = len(visited)
    # Return the remainder when M is divided by 1000
    return M % 1000
result = count_reachable_points()
print(f"\\boxed{{{result}}}")

>>> output
\boxed{373}
```

The answer is $\boxed{373}$

**Example Problem:** The AIME Triathlon consists of a half-mile swim, a 30-mile bicycle ride, and an eight-mile run. Tom swims, bicycles, and runs at constant rates. He runs fives times as fast as he swims, and he bicycles twice as fast as he runs. Tom completes the AIME Triathlon in four and a quarter hours. How many minutes does he spend bicycling?

**Example Solution:**
Let's approach this problem step-by-step:

1. Let's define variables:
   - Let x be Tom's swimming speed in miles per hour
   - Then 5x is his running speed in miles per hour
   - And 10x is his bicycling speed in miles per hour

2. We can set up equations based on the given information:
   - Swimming time: 0.5 / x hours
   - Bicycling time: 30 / (10x) hours
   - Running time: 8 / (5x) hours

3. The total time is 4.25 hours, so we can set up an equation:
   0.5/x + 30/(10x) + 8/(5x) = 4.25

4. Let's solve this equation and find the bicycling time.

Here's the Python code to solve this:

```
from sympy import symbols, Eq, solve, Rational
def solve_triathlon():
    x = symbols('x')
    # Set up the equation
    eq = Eq(Rational(1,2)/x + 30/(10*x) + 8/(5*x), Rational(17,4))
    # Solve the equation
    solution = solve(eq)[0]
    # Calculate bicycling time in hours
    bike_time = 30 / (10 * solution)
    # Convert to minutes
    bike_time_minutes = int(bike_time * 60)
    return f"\\boxed{{{bike_time_minutes}}}"
print(solve_triathlon())

>>> output:
\boxed{150}
```

The answer is $\boxed{150}$

**Example Problem:** Let $S$ be the increasing sequence of positive integers whose binary representation has exactly $8$ ones. Let $N$ be the 1000th number in $S$. Find the remainder when $N$ is divided by 1000

**Example Solution:**
Let's approach this step-by-step:

1. We need to generate the sequence S of numbers with exactly 8 ones in their binary representation.

2. We'll find the 1000th number in this sequence.

3. Finally, we'll calculate the remainder when this number is divided by 1000.

Here's the Python code to solve this:

```python
def count_ones(n):
    return bin(n).count('1')
def solution():
    count = 0
    n = 1
    while count < 1000:
        if count_ones(n) == 8:
            count += 1
            if count == 1000:
                return n % 1000
        n += 1
result = solution()
print(result)

>>> output:
\boxed{32}
```

The answer is $\boxed{32}$

**Example Problem:** Two geometric sequences $a_1, a_2, a_3, \ldots$ and $b_1, b_2, b_3, \ldots$ have the same common ratio, with $a_1 = 27$ $b_1 = 99$ , and $a_{15} = b_{11}$ . Find $a_9$

**Example Solution:**
Let's approach this step-by-step:

1. Let's denote the common ratio as r.

2. We can express a and b in terms of their first terms and r.

3. We can then equate these expressions to find r.

4. Once we have r, we can calculate a.

Here's the Python code to solve this:

```python
from sympy import symbols, Eq, solve, Rational
def find_a9():
    r = symbols('r')
    #   a    =  a   *  r
    #   b    =  b   *  r
    #   a    =  b
    eq = Eq(27 * r**14, 99 * r**10)
    # Solve for r
    solution = solve(eq)
    # We expect only one positive real solution
    r_value = [sol for sol in solution if sol.is_real and sol > 0][0]
    # Calculate  a
    a9 = 27 * r_value**8
    return a9
result = find_a9()
print(f"\\boxed{{{result}}}")

>>> output
\boxed{363}
```

The answer is $\boxed{363}$

**For AMC**

Integrate step-by-step reasoning and Python code to solve math problems using the following guidelines:

- Analyze the question and write functions to solve the problem; the function should not take any arguments. - Present the final result in LaTeX using a '☐' without any units. - Utilize the 'pi' symbol and 'Rational' from Sympy for $\pi$ and fractions, and simplify all fractions and square roots without converting them to decimal values.

Once you get the code output, just display the output answer within '☐' and terminate. Here are some examples you may refer to:

**Example Problem:** Small lights are hung on a string 6 inches apart in the order red, red, green, green, green, red, red, green, green, green, and so on continuing this pattern of 2 red lights followed by 3 green lights. How many feet separate the 3rd red light and the 21st red light? Note: 1 foot is equal to 12 inches.

**Example Solution:**
Let's approach this step-by-step:

1. First, we need to understand the pattern of lights:
   - Every 5 lights (2 red + 3 green) complete one cycle of the pattern.
   - Each cycle is 30 inches long (5 lights * 6 inches).

2. We need to find how many complete cycles are between the 3rd and 21st red lights:
   - 3rd red light is at the end of the 2nd cycle (as each cycle has 2 red lights).
   - 21st red light is at the end of the 11th cycle (as 21 ÷ 2 = 10.5, rounded up to 11).
   - So there are 9 complete cycles between them (11 - 2 = 9).

3. Calculate the distance:
   - 9 cycles * 30 inches per cycle = 270 inches
   - Convert 270 inches to feet: 270 ÷ 12 = 22.5 feet

Here's the Python code to solve this:

```python
from sympy import Rational
def calculate_distance():
    lights_per_cycle = 5
    inches_per_light = 6
    inches_per_cycle = lights_per_cycle * inches_per_light
    start_cycle = 2   # 3rd red light is at the end of 2nd cycle
    end_cycle = 11    # 21st red light is at the end of 11th cycle
    cycles_between = end_cycle - start_cycle
    total_inches = cycles_between * inches_per_cycle
    total_feet = Rational(total_inches, 12)
    return f"\\boxed{{{total_feet}}}"
print(calculate_distance())

>>> output:
\boxed{45/2}
```

The answer is $\boxed{45/2}$

**Example Problem:** A fruit salad consists of blueberries, raspberries, grapes, and cherries. The fruit salad has a total of 280 pieces of fruit. There are twice as many raspberries as blueberries, three times as many grapes as cherries, and four times as many cherries as raspberries. How many cherries are there in the fruit salad?

**Example Solution:**
Let's approach this problem step-by-step:

1. Let's define variables for each fruit:
   b = number of blueberries
   r = number of raspberries
   g = number of grapes
   c = number of cherries

2. We can set up equations based on the given information:
   r = 2b (twice as many raspberries as blueberries)
   g = 3c (three times as many grapes as cherries)
   c = 4r (four times as many cherries as raspberries)

3. We also know that the total number of fruits is 280:
   b + r + g + c = 280

4. Now, let's write a Python function to solve this system of equations:

```python
from sympy import symbols, Eq, solve
def solve_fruit_salad():
    b, r, g, c = symbols('b r g c')
    eq1 = Eq(r, 2*b)
    eq2 = Eq(g, 3*c)
    eq3 = Eq(c, 4*r)
    eq4 = Eq(b + r + g + c, 280)
    solution = solve((eq1, eq2, eq3, eq4))
    return f"\\boxed{{{solution[c]}}}"
print(solve_fruit_salad())

>>> output:
\boxed{64}
```

The answer is $\boxed{64}$

**Example Problem:** Last summer $30\%$ of the birds living on Town Lake were geese, $25\%$ were swans, $10\%$ were herons, and $35\%$ were ducks. What percent of the birds that were not swans were geese?

**Example Solution:**
Let's approach this step-by-step:

1. First, we need to calculate the percentage of birds that were not swans.

2. Then, we'll calculate what percentage of this non-swan group were geese.

Here's the Python code to solve this:

```python
from sympy import Rational
def calculate_geese_percentage():
    # Total percentage of birds
    total = 100
    # Percentage of swans
    swans = 25
    # Percentage of non-swans
    non_swans = total - swans
    # Percentage of geese among all birds
    geese = 30
    # Calculate the percentage of geese among non-swans
    geese_among_non_swans = Rational(geese, non_swans) * 100
    # Simplify the fraction
    result = geese_among_non_swans.simplify()
    return f"\\boxed{{{result}}}"
print(calculate_geese_percentage())

>>> output
\boxed{40}
```

The answer is $\boxed{40}$

**Example Problem**: At a twins and triplets convention, there were 9 sets of twins and 6 sets of triplets, all from different families. Each twin shook hands with all the twins except his/her siblings and with half the triplets. Each triplet shook hands with all the triplets except his/her siblings and with half the twins. How many handshakes took place?

**Example Solution:**
Let's approach this problem step-by-step:

1. Calculate the number of handshakes between twins

2. Calculate the number of handshakes between triplets

3. Calculate the number of handshakes between twins and triplets

4. Sum up all handshakes

Here's the Python code to solve this:

```python
from sympy import Rational
def calculate_handshakes():
    # Number of twins and triplets
    twins = 9 * 2
    triplets = 6 * 3
    # Handshakes between twins
    twin_handshakes = (twins * (twins - 2)) // 2  # Each twin shakes
    hands with all but 1 other twin
    # Handshakes between triplets
    triplet_handshakes = (triplets * (triplets - 3)) // 2  # Each triplet
     shakes hands with all but 2 other triplets
    # Handshakes between twins and triplets
    twin_triplet_handshakes = twins * triplets * Rational(1, 2)  # Each
    twin shakes hands with half the triplets
    # Total handshakes
    total_handshakes = twin_handshakes + triplet_handshakes +
    twin_triplet_handshakes
    return f"\\boxed{{{int(total_handshakes)}}}"
print(calculate_handshakes())

>>> output
\boxed{441}
```

The answer is $\boxed{441}$

### A.7    SBSC System-Prompt Tuning

For few-shot learning, apart from relevant exemplars, the LLM also benefits from a general instruction at the beginning [38, 13, 28] that provides a guideline or context about how the model should approach the task, particularly those requiring logical reasoning, multi-step operations, etc. This can be specially useful when the task requires a more nuanced understanding and when the instructions need to be followed rigorously, as is the case with SBSC.

The following has been used for our method:
You are given a math problem and you need to think of a stepwise approach/process to be followed to solve the problem. Use sympy-based python code to codify each of these steps in sequential manner. You must end each of your code snippet with a print statement followed by *output* in the next line.
**Use the results from the output of the code snippets of each step and continue to the next step until you finish the final step and solve the problem completely.** **1**
**In each new code block, you must define the variables and symbols again based on the previous code snippets or their outputs as suitable.** **2**
Use sympy in appropriate way, as and when required.
Once you complete the last step, write this line: ### END OF CODE
Then just print the final numeric answer (without any units) like this: The final answer is boxed{}.
**If the executed code snippet returns an error, use it to correct the current step's code snippet. DO NOT restart solving from Step 1.** **3**

In Table 3, we compare the importance of the highlighted lines 1,2 and 3 above by trying out different combinations with/without them. We find that the complete instruction seems to be the best and proceed to use it for all experiments.

| | Full | 3 only | 2+3 | 1+3 |
|---|---|---|---|---|
| AMC 3 Yrs | 48 | 43 | 49 | 46 |
| AIME 3 Yrs | 29 | 24 | 28 | 25 |

Table 3: Comparison of GPT-4o scores across different variations of prompt instructions.

## A.8 SBSC Prompt for AMC

You are given a math problem and you need to think of a stepwise approach/process to be followed to solve the problem. Use sympy-based python code to codify each of these steps in sequential manner. You must end each of your code snippet with a print statement followed by *output* in the next line. Use the results from the output of the code snippets of each step and continue to the next step until you finish the final step and solve the problem completely.

In each new code block, you must define the variables and symbols again based on the previous code snippets or their outputs as suitable.

Use sympy in appropriate way, as and when required.

Once you complete the last step, write this line: ### END OF CODE

Then just print the final numeric answer (without any units) like this: The final answer is boxed{}.

If the executed code snippet returns an error, use it to correct the current step's code snippet. DO NOT restart solving from Step 1.

Here are a few examples for your reference.

**Example Problem:** Small lights are hung on a string 6 inches apart in the order red, red, green, green, green, red, red, green, green, green, and so on continuing this pattern of 2 red lights followed by 3 green lights. How many feet separate the 3rd red light and the 21st red light? Note: 1 foot is equal to 12 inches.

**Example Solution:** Let's solve this problem step by step using SymPy-based Python code.

Step 1: Define the pattern and calculate the number of lights in one complete pattern

```python
from sympy import symbols, Rational
# Define the number of red and green lights in one pattern
red_lights_per_pattern = 2
green_lights_per_pattern = 3
# Calculate total lights in one pattern
lights_per_pattern = red_lights_per_pattern + green_lights_per_pattern
print(f"Number of lights in one complete pattern: {lights_per_pattern}")

>>> output:
Number of lights in one complete pattern: 5
```

Let's continue with the next steps to solve this problem and leveraging the outputs of previous steps.
Step 2: Calculate the number of complete patterns between the 3rd and 21st red lights

```python
from sympy import symbols, Rational
# Define the number of red lights we're considering
start_red_light = 3
end_red_light = 21
# Calculate the number of red lights between the 3rd and 21st (inclusive)
red_lights_between = end_red_light - start_red_light + 1
# Calculate the number of complete patterns
complete_patterns = (red_lights_between - 1) // 2
print(f"Number of complete patterns between the 3rd and 21st red lights:
    {complete_patterns}")

>>> output:
Number of complete patterns between the 3rd and 21st red lights: 9
```

Let's continue with the next steps to solve this problem and leveraging the outputs of previous steps.
Step 3: Calculate the total number of lights between the 3rd and 21st red lights

```python
from sympy import symbols, Rational
# Define the number of red and green lights in one pattern
red_lights_per_pattern = 2
green_lights_per_pattern = 3
lights_per_pattern = red_lights_per_pattern + green_lights_per_pattern
# Number of complete patterns
complete_patterns = 9
# Calculate total lights
total_lights = complete_patterns * lights_per_pattern + 1  # +1 for the
    21st red light
print(f"Total number of lights between the 3rd and 21st red lights (
    inclusive): {total_lights}")

>>> output:
Total number of lights between the 3rd and 21st red lights (inclusive):
    46
```

Let's continue with the next steps to solve this problem and leveraging the outputs of previous steps.
Step 4: Calculate the total distance in inches

```python
from sympy import symbols, Rational
# Total number of lights
total_lights = 46
# Distance between lights in inches
distance_between_lights = 6
# Calculate total distance in inches
total_distance_inches = (total_lights - 1) * distance_between_lights
print(f"Total distance in inches: {total_distance_inches}")

>>> output:
Total distance in inches: 270
```

Let's continue with the next steps to solve this problem and leveraging the outputs of previous steps.
Step 5: Convert the distance from inches to feet

```python
from sympy import symbols, Rational
# Total distance in inches
total_distance_inches = 270
# Convert inches to feet
inches_per_foot = 12
total_distance_feet = Rational(total_distance_inches, inches_per_foot)
print(f"Total distance in feet: {total_distance_feet}")

>>> output:
Total distance in feet: 45/2
```

### END OF CODE

The final answer is $\boxed{\dfrac{45}{2}}$

**Example Problem:** A fruit salad consists of blueberries, raspberries, grapes, and cherries. The fruit salad has a total of 280 pieces of fruit. There are twice as many raspberries as blueberries, three times as many grapes as cherries, and four times as many cherries as raspberries. How many cherries are there in the fruit salad?

**Example Solution:** Let's solve this problem step by step using SymPy-based Python code.
Step 1: Define the variables and set up the equations

```python
from sympy import symbols, Eq, solve
# Define symbols for each type of fruit
b, r, g, c = symbols('b r g c')
# Set up equations based on the given information
eq1 = Eq(b + r + g + c, 280)  # Total pieces of fruit
eq2 = Eq(r, 2*b)   # Twice as many raspberries as blueberries
eq3 = Eq(g, 3*c)   # Three times as many grapes as cherries
```

```
eq4 = Eq(c, 4*r)   # Four times as many cherries as raspberries
print("Equations:")
print(eq1)
print(eq2)
print(eq3)
print(eq4)

>>> output:
Equations:
Eq(b + c + g + r, 280)
Eq(r, 2*b)
Eq(g, 3*c)
Eq(c, 4*r)
```

Let's continue with the next steps to solve this problem and leveraging the outputs of previous steps.
Step 2: Solve the system of equations

```
from sympy import symbols, Eq, solve
# Define symbols for each type of fruit
b, r, g, c = symbols('b r g c')
# Set up equations based on the given information
eq1 = Eq(b + r + g + c, 280)  # Total pieces of fruit
eq2 = Eq(r, 2*b)   # Twice as many raspberries as blueberries
eq3 = Eq(g, 3*c)   # Three times as many grapes as cherries
eq4 = Eq(c, 4*r)   # Four times as many cherries as raspberries
# Solve the system of equations
solution = solve((eq1, eq2, eq3, eq4))
print("Solution:")
print(solution)

>>> output:
Solution:
{b: 8, c: 64, g: 192, r: 16}
```

Let's continue with the next steps to solve this problem and leveraging the outputs of previous steps.
Step 3: Extract the number of cherries from the solution

```
from sympy import symbols, Eq, solve
# Define symbols for each type of fruit
b, r, g, c = symbols('b r g c')
# Set up equations based on the given information
eq1 = Eq(b + r + g + c, 280)  # Total pieces of fruit
eq2 = Eq(r, 2*b)   # Twice as many raspberries as blueberries
eq3 = Eq(g, 3*c)   # Three times as many grapes as cherries
eq4 = Eq(c, 4*r)   # Four times as many cherries as raspberries
# Solve the system of equations
solution = solve((eq1, eq2, eq3, eq4))
# Extract the number of cherries
cherries = solution[c]
print(f"The number of cherries in the fruit salad is: {cherries}")

>>> output:
The number of cherries in the fruit salad is: 64
```

END OF CODE The final answer is $\boxed{64}$

**Example Problem:** Last summer $30\%$ of the birds living on Town Lake were geese, $25\%$ were swans, $10\%$ were herons, and $35\%$ were ducks. What percent of the birds that were not swans were geese?

**Example Solution:** Let's solve this problem step by step using SymPy-based Python code.

Step 1: Define the percentages of each type of bird

```
from sympy import symbols, Rational
geese = Rational(30, 100)
```

```
swans = Rational(25, 100)
herons = Rational(10, 100)
ducks = Rational(35, 100)
print(f"Percentage of geese: {geese}")
print(f"Percentage of swans: {swans}")
print(f"Percentage of herons: {herons}")
print(f"Percentage of ducks: {ducks}")

>>> output:
Percentage of geese: 3/10
Percentage of swans: 1/4
Percentage of herons: 1/10
Percentage of ducks: 7/20
```

Let's continue with the next steps to solve this problem. Step 2: Calculate the percentage of birds that are not swans

```
from sympy import symbols, Rational
geese = Rational(30, 100)
swans = Rational(25, 100)
herons = Rational(10, 100)
ducks = Rational(35, 100)
not_swans = 1 - swans
print(f"Percentage of birds that are not swans: {not_swans}")

>>> output:
Percentage of birds that are not swans: 3/4
```

Let's continue with the next steps to solve this problem and leveraging the outputs of previous steps. Step 3: Calculate the percentage of geese among the birds that are not swans

```
from sympy import symbols, Rational
geese = Rational(30, 100)
swans = Rational(25, 100)
herons = Rational(10, 100)
ducks = Rational(35, 100)
not_swans = 1 - swans
geese_among_not_swans = geese / not_swans
print(f"Percentage of geese among birds that are not swans: {
    geese_among_not_swans}")

>>> output:
Percentage of geese among birds that are not swans: 2/5
```

Let's continue with the next steps to solve this problem and leveraging the outputs of previous steps. Step 4: Convert the fraction to a percentage

```
from sympy import symbols, Rational
geese = Rational(30, 100)
swans = Rational(25, 100)
herons = Rational(10, 100)
ducks = Rational(35, 100)
not_swans = 1 - swans
geese_among_not_swans = geese / not_swans
percentage = geese_among_not_swans * 100
print(f"Percentage of geese among birds that are not swans: {percentage}%
    ")

>>> output:
Percentage of geese among birds that are not swans: 40%
```

### END OF CODE

The final answer is $\boxed{40}$

Example Problem: At a twins and triplets convention, there were 9 sets of twins and 6 sets of triplets, all from different families. Each twin shook hands with all the twins except his/her siblings and with half the triplets. Each triplet shook hands with all the triplets except his/her siblings and with half the twins. How many handshakes took place?

**Example Solution:** Let's solve this problem step by step using SymPy-based Python code.

Step 1: Define the given information

```python
from sympy import symbols, Rational
# Number of sets of twins and triplets
twin_sets = 9
triplet_sets = 6
# Number of individuals
twins = twin_sets * 2
triplets = triplet_sets * 3
print(f"Number of twins: {twins}")
print(f"Number of triplets: {triplets}")

>>> output:
Number of twins: 18
Number of triplets: 18
```

Let's continue with the next steps to solve this problem and leveraging the outputs of previous steps.
Step 2: Calculate handshakes between twins

```python
from sympy import symbols, Rational
# Number of sets of twins and triplets
twin_sets = 9
triplet_sets = 6
# Number of individuals
twins = twin_sets * 2
triplets = triplet_sets * 3
# Handshakes between twins
twin_handshakes = (twins * (twins - 2)) // 2  # Each twin shakes hands
    with all twins except self and sibling
print(f"Handshakes between twins: {twin_handshakes}")

>>> output
Handshakes between twins: 144
```

Let's continue with the next steps to solve this problem and leveraging the outputs of previous steps.
Step 3: Calculate handshakes between triplets

```python
from sympy import symbols, Rational
# Number of sets of twins and triplets
twin_sets = 9
triplet_sets = 6
# Number of individuals
twins = twin_sets * 2
triplets = triplet_sets * 3
# Handshakes between twins
twin_handshakes = (twins * (twins - 2)) // 2
# Handshakes between triplets
triplet_handshakes = (triplets * (triplets - 3)) // 2  # Each triplet
    shakes hands with all triplets except self and two siblings
print(f"Handshakes between triplets: {triplet_handshakes}")

>>> output
Handshakes between triplets: 135
```

Let's continue with the next steps to solve this problem and leveraging the outputs of previous steps.
Step 4: Calculate handshakes between twins and triplets

```python
from sympy import symbols, Rational
```

```
# Number of sets of twins and triplets
twin_sets = 9
triplet_sets = 6
# Number of individuals
twins = twin_sets * 2
triplets = triplet_sets * 3
# Handshakes between twins
twin_handshakes = (twins * (twins - 2)) // 2
# Handshakes between triplets
triplet_handshakes = (triplets * (triplets - 3)) // 2
# Handshakes between twins and triplets
twin_triplet_handshakes = twins * (triplets // 2)  # Each twin shakes
    hands with half the triplets
print(f"Handshakes between twins and triplets: {twin_triplet_handshakes}"
    )

>>> output:
Handshakes between twins and triplets: 162
```

Let's continue with the next steps to solve this problem and leveraging the outputs of previous steps.
Step 5: Calculate the total number of handshakes

```
from sympy import symbols, Rational
# Number of sets of twins and triplets
twin_sets = 9
triplet_sets = 6
# Number of individuals
twins = twin_sets * 2
triplets = triplet_sets * 3
# Handshakes between twins
twin_handshakes = (twins * (twins - 2)) // 2
# Handshakes between triplets
triplet_handshakes = (triplets * (triplets - 3)) // 2
# Handshakes between twins and triplets
twin_triplet_handshakes = twins * (triplets // 2)
# Total handshakes
total_handshakes = twin_handshakes + triplet_handshakes +
    twin_triplet_handshakes
print(f"Total number of handshakes: {total_handshakes}")

>>> output
Total number of handshakes: 441
```

### END OF CODE

The final answer is $\boxed{441}$

## A.9   SBSC Prompt for AIME

You are given a math problem and you need to think of a stepwise approach/process to be followed to solve the problem. Use sympy-based python code to codify each of these steps in sequential manner. You must end each of your code snippet with a print statement followed by *output* in the next line. Use the results from the output of the code snippets of each step and continue to the next step until you finish the final step and solve the problem completely.
In each new code block, you must define the variables and symbols again based on the previous code snippets or their outputs as suitable.
Use sympy in appropriate way, as and when required.
Once you complete the last step, write this line: ### END OF CODE
Then just print the final numeric answer (without any units) like this: The final answer is boxed{}.
If the executed code snippet returns an error, use it to correct the current step's code snippet. DO NOT restart solving from Step 1.

Here are a few examples for your reference.

**Example Problem:** A frog begins at $P_0 = (0,0)$ and makes a sequence of jumps according to the following rule: from $P_n = (x_n, y_n)$, the frog jumps to $P_{n+1}$, which may be any of the points $(x_n + 7, y_n + 2)$, $(x_n + 2, y_n + 7)$, $(x_n - 5, y_n - 10)$, or $(x_n - 10, y_n - 5)$. There are $M$ points $(x, y)$ with $|x| + |y| \leq 100$ that can be reached by a sequence of such jumps. Find the remainder when $M$ is divided by 1000.

**Example Solution:** Let's approach this problem step by step using SymPy-based Python code.

Step 1: Import necessary libraries and define the jump rules

```
from sympy import symbols, Abs
import numpy as np
# Define the possible jumps from any position (x, y)
def jumps(x, y):
    return [
        (x + 7, y + 2),    # Jump by (7, 2)
        (x + 2, y + 7),    # Jump by (2, 7)
        (x - 5, y - 10),   # Jump by (-5, -10)
        (x - 10, y - 5)    # Jump by (-10, -5)
    ]
print("Jump rules defined.")

>>> output
Jump rules defined.
```

Let's continue with the next steps to solve this problem and leveraging the outputs of previous steps.
Step 2: Implement a breadth-first search to find all reachable points

```
from sympy import symbols, Abs
import numpy as np
from collections import deque
# Reuse the jumps function to define possible moves
def jumps(x, y):
    return [
        (x + 7, y + 2),
        (x + 2, y + 7),
        (x - 5, y - 10),
        (x - 10, y - 5)
    ]
# Define the breadth-first search (BFS) function
def bfs():
    queue = deque([(0, 0)])  # Start BFS from the point (0, 0)
    visited = set([(0, 0)])  # Keep track of visited points to avoid
    reprocessing
    # Process each point in the queue
    while queue:
        x, y = queue.popleft()
        for nx, ny in jumps(x, y):  # Try all possible jumps from the
    current point
            if (nx, ny) not in visited and abs(nx) + abs(ny) <= 100:
                visited.add((nx, ny))   # Mark the new point as visited
                queue.append((nx, ny))   # Add the new point to the queue
    to explore further
    return len(visited)   y
M = bfs()
print(f"Number of reachable points (M): {M}")

>>> output:
Number of reachable points (M): 1373
```

Let's continue with the final step to solve this problem. Step 3: Calculate the remainder when M is divided by 1000

```
from sympy import symbols, Abs
import numpy as np
M = 1373  # M_value from previous step
```

```
# Find the remainder when M is divided by 1000
remainder = M % 1000
print(f"Remainder when M is divided by 1000: {remainder}")

>>> output
Remainder when M is divided by 1000: 373
```

### END OF CODE

The final answer is $\boxed{373}$

**Example Problem:** The AIME Triathlon consists of a half-mile swim, a 30-mile bicycle ride, and an eight-mile run. Tom swims, bicycles, and runs at constant rates. He runs fives times as fast as he swims, and he bicycles twice as fast as he runs. Tom completes the AIME Triathlon in four and a quarter hours. How many minutes does he spend bicycling?

**Example Solution:** Let's solve this problem step by step using SymPy-based Python code.

Step 1: Define the variables and set up the basic equations

```
from sympy import symbols, Eq, solve
# Define variables
s, b, r = symbols('s b r')  # s: swim speed, b: bike speed, r: run speed
# Set up equations based on the given information
eq1 = Eq(r, 5*s)  # Tom runs 5 times as fast as he swims
eq2 = Eq(b, 2*r)  # Tom bicycles twice as fast as he runs
print("Equation 1:", eq1)
print("Equation 2:", eq2)

>>> output
Equation 1: Eq(r, 5*s)
Equation 2: Eq(b, 2*r)
```

Let's continue with the next steps to solve this problem and leveraging the outputs of previous steps.
Step 2: Set up the equation for the total time

```
from sympy import symbols, Eq, solve
# Reuse earlier code
s, b, r = symbols('s b r')
eq1 = Eq(r, 5*s)
eq2 = Eq(b, 2*r)
# Total time equation (in hours)
total_time = 0.5/s + 30/b + 8/r  #The AIME Triathlon consists of a half-
    mile swim, a 30-mile bicycle ride, and an eight-mile run.
eq3 = Eq(total_time, 4.25)  # Total time is 4.25 hours
print("Total time equation:", eq3)

>>> output:
Total time equation: Eq(0.5/s + 8/r + 30/b, 4.25)
```

Let's continue with the next steps to solve this problem and leveraging the outputs of previous steps.
Step 3: Substitute the relationships between speeds into the total time equation

```
from sympy import symbols, Eq, solve, simplify
# Resue earlier code
s, b, r = symbols('s b r')
eq1 = Eq(r, 5*s)
eq2 = Eq(b, 2*r)
total_time = 0.5/s + 30/b + 8/r
eq3 = Eq(total_time, 4.25)  # Total time is 4.25 hours
# Substitute r and b in terms of s to simplify the equation
eq3_substituted = eq3.subs([(r, 5*s), (b, 10*s)])
eq3_simplified = simplify(eq3_substituted)
print("Simplified equation:", eq3_simplified)
```

```
>>> output:
Simplified equation: Eq(5.1/s, 4.25)
```

Let's continue with the next steps to solve this problem and leveraging the outputs of previous steps.
Step 4: Solve for the swim speed (s)

```python
from sympy import symbols, Eq, solve, simplify
# Redefine variables
s, b, r = symbols('s b r')  # s: swim speed, b: bike speed, r: run speed
# Simplified equation from previous step
eq3_simplified = Eq(5.1/s, 4.25)
# Solve for s
s_solution = solve(eq3_simplified, s)
print("Swim speed (s):", s_solution[0])

>>> output
Swim speed (s): 1.20000000000000
```

Let's continue with the next steps to solve this problem and leveraging the outputs of previous steps.
Step 5: Calculate the bike speed (b)

```python
from sympy import symbols, Eq, solve, simplify
# Redefine variables
s, b, r = symbols('s b r')  # s: swim speed, b: bike speed, r: run speed
# Swim speed from previous step
s_value = 1.20000000000000
# Calculate bike speed
b_value = 10 * s_value  # b = 2r = 2(5s) = 10s
print("Bike speed (b):", b_value)

>>> output
Bike speed (b): 12.0
```

Let's continue with the next steps to solve this problem and leveraging the outputs of previous steps.
Step 6: Calculate the time spent bicycling

```python
from sympy import symbols, Eq, solve, simplify
# Redefine variables
s, b, r = symbols('s b r')  # s: swim speed, b: bike speed, r: run speed
# Bike speed from previous step
b_value = 12.0
# Calculate time spent bicycling
bike_distance = 30  # miles
bike_time = bike_distance / b_value  # hours
# Convert to minutes
bike_time_minutes = bike_time * 60
print("Time spent bicycling (in minutes):", bike_time_minutes)

>>> output
Time spent bicycling (in minutes): 150.0
```

### END OF CODE

The final answer is $\boxed{150}$

**Example Problem:** Let $S$ be the increasing sequence of positive integers whose binary representation has exactly 8 ones. Let $N$ be the 1000th number in $S$. Find the remainder when $N$ is divided by 1000

**Example Solution:** Let's solve this problem step by step using SymPy-based Python code.

Step 1: Import necessary libraries and define a function to count ones in binary representation

```python
from sympy import symbols, binomial
import math
# Function to count the number of '1's in the binary representation of a
    number
```

```
def count_ones(n):
    return bin(n).count('1')  # Convert n to binary and count occurrences
     of '1'
print("Function to count ones in binary representation defined.")

>>> output:
Function to count ones in binary representation defined.
```

Let's continue with the next steps to solve this problem and leveraging the outputs of previous steps.
Step 2: Find the smallest number in the sequence S

```
from sympy import symbols, binomial
import math
# Reuse the function to count the number of '1's in binary
def count_ones(n):
    return bin(n).count('1')
# The smallest number in the sequence S is represented by 8 consecutive
    '1's in binary
smallest_number = int('1' * 8, 2)  # Convert '11111111' (binary) to
    decimal
print(f"The smallest number in sequence S is: {smallest_number}")

>>> output
The smallest number in sequence S is: 255
```

Let's continue with the next steps to solve this problem and leveraging the outputs of previous steps.
Step 3: Find the largest number in the sequence S

```
from sympy import symbols, binomial
import math
# Reuse the function to count the number of '1's in binary
def count_ones(n):
    return bin(n).count('1')
smallest_number = int('1' * 8, 2)
# The largest number in sequence S is represented by 8 consecutive '1's
    followed by 24 '0's
largest_number = int('1' * 8 + '0' * 24, 2)  # Convert
    '11111111000000000000000000000000' to decimal
print(f"The largest number in sequence S is: {largest_number}")

>>> output
The largest number in sequence S is: 4278190080
```

Let's continue with the next steps to solve this problem and leveraging the outputs of previous steps.
Step 4: Calculate the total number of elements in sequence S

```
from sympy import symbols, binomial
import math
def count_ones(n):
    return bin(n).count('1')
smallest_number = int('1' * 8, 2)
largest_number = int('1' * 8 + '0' * 24, 2)
# Calculate the total number of elements in sequence S using binomial(32,
     8)
# This represents the number of ways to choose 8 positions out of 32 bits
     to set to '1'
total_elements = binomial(32, 8)
print(f"The total number of elements in sequence S is: {total_elements}")

>>> output
The total number of elements in sequence S is: 10518300
```

Let's continue with the next steps to solve this problem and leveraging the outputs of previous steps.
Step 5: Find the 1000th number in sequence S

```
from sympy import symbols, binomial
import math
#Reuse all earlier code
def count_ones(n):
    return bin(n).count('1')
smallest_number = int('1' * 8, 2)
largest_number = int('1' * 8 + '0' * 24, 2)
total_elements = binomial(32, 8)
# Find the 1000th number in the sequence
count = 0  # Initialize a counter for the sequence
N = smallest_number  # Start from the smallest number in S
# Iterate through numbers and check for 8 '1's in the binary
    representation
while count < 1000:
    if count_ones(N) == 8:  # If the current number has exactly 8 '1's
        count += 1  # Increment the count
    if count == 1000:  # Stop if we've found the 1000th number
        break
    N += 1  # Move to the next number
print(f"The 1000th number (N) in sequence S is: {N}")

>>> output
The 1000th number (N) in sequence S is: 7032
```

Let's continue with the next steps to solve this problem and leveraging the outputs of previous steps.
Step 6: Calculate the remainder when N is divided by 1000

```
from sympy import symbols, binomial
import math
def count_ones(n):
    return bin(n).count('1')
N = 7032  # From the previous step
# Calculate the remainder when N is divided by 1000
remainder = N % 1000
print(f"The remainder when N is divided by 1000 is: {remainder}")

>>> output
The remainder when N is divided by 1000 is: 32
```

### END OF CODE

The final answer is $\boxed{32}$

**Example Problem:** Two geometric sequences $a_1, a_2, a_3, \ldots$ and $b_1, b_2, b_3, \ldots$ have the same common ratio, with $a_1 = 27\ b_1 = 99$, and $a_{15} = b_{11}$. Find $a_9$

**Example Solution:** Let's solve this problem step by step using SymPy-based Python code.

Step 1: Define the variables and set up the initial conditions

```
from sympy import symbols, Eq, solve
# Define symbols
r = symbols('r')  # common ratio
a1, b1 = 27, 99  # first terms of sequences a and b
print(f"a1 = {a1}, b1 = {b1}")
print("Common ratio is represented by r")

>>> output
a1 = 27, b1 = 99
Common ratio is represented by r
```

Let's continue with the next steps to solve this problem and leveraging the outputs of previous steps.
Step 2: Set up the equation for a15 = b11

```
from sympy import symbols, Eq, solve
# Redefine symbols
r = symbols('r')
```

```
a1, b1 = 27, 99
# Set up the equation a15 = b11
eq = Eq(a1 * r**14, b1 * r**10)
print("Equation: a15 = b11")
print(f"Symbolically: {eq}")

>>> output
Equation: a15 = b11
Symbolically: Eq(27*r**14, 99*r**10)
```

Let's continue with the next steps to solve this problem and leveraging the outputs of previous steps.
Step 3: Solve the equation for r

```
from sympy import symbols, Eq, solve
# Redefine symbols
r = symbols('r')
a1, b1 = 27, 99
# Set up the equation a15 = b11
eq = Eq(a1 * r**14, b1 * r**10)
# Solve for r
solution = solve(eq, r)
print("Solutions for r:")
print(solution)

>>> output
Solutions for r:
[0, -11**(1/4)*3**(3/4)/3, 11**(1/4)*3**(3/4)/3]
```

Let's continue with the next steps to solve this problem and leveraging the outputs of previous steps.
Step 4: Identify the correct value for r

```
from sympy import symbols, Eq, solve, N
# Reuse earlier code
r = symbols('r')
a1, b1 = 27, 99
eq = Eq(a1 * r**14, b1 * r**10)
solution = solve(eq, r)
# The correct value for r is the positive real solution
r_value = max(sol for sol in solution if sol.is_real)
print(f"The common ratio r is: {N(r_value, 6)}")

>>> output
The common ratio r is: 1.38378
```

Let's continue with the next steps to solve this problem and leveraging the outputs of previous steps.
Step 5: Calculate a9

```
from sympy import symbols, Eq, solve, N
# Reuse earlier code
r = symbols('r')
a1, b1 = 27, 99
eq = Eq(a1 * r**14, b1 * r**10)
solution = solve(eq, r)
r_value = max(sol for sol in solution if sol.is_real)
# Calculate a9
a9 = a1 * r_value**8
print(f"a9 = {N(a9, 10)}")

>>> output
a9 = 363.0000000
```

### END OF CODE

The final answer is $\boxed{363}$

## A.10  Results with Standard Deviations

| Method | AMC | | AIME | | MathOdyssey | |
|---|---|---|---|---|---|---|
| | greedy | maj@7 | greedy | maj@7 | greedy | maj@7 |
| Claude-3.5-Sonnet | | | | | | |
| COT | 31.16 ($\pm$1.0) | 35.79 | 9.09 ($\pm$1.0) | 10.91 | 11.89 ($\pm$0.6) | 16.89 |
| PAL | 35.79 ($\pm$1.0) | 36.42 | $\underline{27.48}$ ($\pm$0.6) | $\underline{28.79}$ | 27.23 ($\pm$0.6) | 31.01 |
| TIR-ToRA | $\underline{38.59}$ ($\pm$0.6) | $\underline{43.16}$ | 24.64 ($\pm$3.2) | 26.67 | $\underline{27.23}$ ($\pm$0.6) | $\underline{32.43}$ |
| SBSC (Ours) | **49.33** ($\pm$3.1)$_{\uparrow 10.7}$ | $-_{\uparrow 6.2}$ | **35.45** ($\pm$1.7)$_{\uparrow 8}$ | $-_{\uparrow 6.7}$ | **39.86** ($\pm$1.0)$_{\uparrow 12.6}$ | $-_{\uparrow 7.4}$ |
| GPT-4o | | | | | | |
| COT | 35.94 ($\pm$0.6) | 37.47 | 10.39 ($\pm$2.1) | 12.12 | 13.51 ($\pm$1.0) | 17.57 |
| PAL | 36.48 ($\pm$0.6) | 38.11 | $\underline{24.63}$ ($\pm$0.6) | $\underline{26.97}$ | 15.74 ($\pm$0.6) | 20.27 |
| TIR-ToRA | $\underline{37.33}$ ($\pm$2.5) | $\underline{40.42}$ | 22.42 ($\pm$1.7) | 25.45 | $\underline{19.59}$ ($\pm$2.6) | $\underline{23.64}$ |
| SBSC (Ours) | **44.55** ($\pm$0.6)$_{\uparrow 7.2}$ | $-_{\uparrow 4.1}$ | **30.7** ($\pm$1.1)$_{\uparrow 6.1}$ | $-_{\uparrow 3.7}$ | **26.55** ($\pm$1.1)$_{\uparrow 7}$ | $-_{\uparrow 2.9}$ |

Table 4: Benchmarking SBSC against different math reasoning methods across three datasets. We report average accuracy over 3 runs with standard deviation within parentheses. Best result in each setting is highlighted in **bold** and second best is underlined. Absolute improvement in performance by SBSC over the previous best method in each setting is indicated in subscript.

