# OpenReview forum: "SBSC: Step-by-Step Coding for Improving Mathematical Olympiad Performance"
_NeurIPS.cc/2024/Workshop/MATH-AI — MATH-AI 24_

### Official Review · Reviewer_ymT6 · 2024-10-06
**A novel step-by-step framework for LLMs in solving complex mathematical problems**

**Rating:** 8
**Confidence:** 3

**Review:**

The paper proposes a framework called Step-by-Step Coding (SBSC) for solving complex mathematical problems using large language models.

The authors conduct extensive experiments comparing SBSC against other state-of-the-art methods like Chain-of-Thought (COT), Program-Aided Language Models (PAL), and Tool-Integrated Reasoning Agent (TIR-ToRA) on datasets like AIME, AMC, and MathOdyssey.

The results show significant improvements in performance using SBSC, with absolute gains of 6-12% over existing methods across different datasets and models (Claude-3.5-Sonnet and GPT-4o).

The paper provides detailed ablation studies to analyze the impact of different components of SBSC.

One of the weakness is that the evaluation is limited to text-based integer-answer questions. Though it is fair to consider only text-based questions, limiting to integer-answer only questions make the evaluation less general.

---

### Official Review · Reviewer_VHLo · 2024-10-07
**A good approach to solve complex olympiad-level reasoning problems**

**Rating:** 7
**Confidence:** 4

**Review:**

**Summary**

The authors propose a LLM-based multi-turn reasoning framework for solving mathematical olympiad-level problems. They develop this Step-by-Step Coding (SBSC) framework to generate a sequence of programs to solve a series of sub-tasks related to the main problem statement leading to a more controlled and granular approach to problem-solving.
The framework tries to leverage the program-generation and in-context learning capabilities of the current state-of-the-art LLMs.
The framework, at each turn, generates an intermediate sub-task along with a program to solve that conditioned on the outputs of the previous turns, initial prompt and given question.


**Strengths**

The authors have explained the pipeline clearly with some good examples and their analysis as to why SBSC could be performing better than the other approaches that try to solve the problem in one-shot program generation.
This work tries to emphasize that complex reasoning tasks might be well-solved if broken down into smaller subproblems/subtasks as they might require paying some extra attention to specific constraints, edge-cases or might require some further simplification.
SBSC is shown to be better at rectifying particular steps in case of errors/erroneous outputs.
This framework can be extended to other tasks as well that might require complex reasoning and/or require focus on certain not-so-common constraints.


**Weaknesses**

How did the authors come up with 6-7 turns per problem?


**Limitations**

The authors did mention that their framework is able to handle only non-image based problems with integer-based answers only.

---

### Official Review · Reviewer_175Q · 2024-10-07

**Rating:** 7
**Confidence:** 4

**Review:**

## Paper summary
The paper introduces Step-by-Step Coding (SBSC), a framework aimed at improving LLM performance on Olympiad-level math problems by using a multi-turn approach where each sub-task is addressed through code generation and execution.

## Strength
- The method is easy to understand and technically sound.
- The method leads to performance improvement over popular methods like Chain-of-Thought (COT), PAL, and TIR-ToRA on some challenging datasets.

## Areas to improve
- The comparisons should also include efficiency (e.g., in terms of API cost or number of tokens).
- Clarity:
  - Why are SBSC's `maj@7` results missing in Table 1?
  - Please explicitly explain AMC3Yrs and AIME3Yrs in Figure 1.
  - Line 20: "elementary GSM8k [9] high-school level MATH [14]." $\to$ "elementary GSM8k [9] \& high-school level MATH [14]."

---

### Decision · Program_Chairs · 2024-10-09

Accept